# Functional Near-Infrared Spectroscopy for the Classification of Motor-Related Brain Activity on the Sensor-Level

**DOI:** 10.3390/s20082362

**Published:** 2020-04-21

**Authors:** Alexander E. Hramov, Vadim Grubov, Artem Badarin, Vladimir A. Maksimenko, Alexander N. Pisarchik

**Affiliations:** 1Neuroscience and Cognitive Technology Laboratory, Center for Technologies in Robotics and Mechatronics Components, Innopolis University, Universitetskaja Str., 1, 420500 Innopolis, Russia; v.grubov@innopolis.ru (V.G.); a.badarin@innopolis.ru (A.B.); v.maksimenko@innopolis.ru (V.A.M.); alexander.pisarchik@ctb.upm.es (A.N.P.); 2Saratov State Medical University, Bolshaya Kazachya Str., 112, 410012 Saratov, Russia; 3Center for Biomedical Technology, Technical University of Madrid, Campus Montegancedo, Pozuelo de Alarcón, 28223 Madrid, Spain

**Keywords:** brain activity, functional near-infrared spectroscopy (fNIRS), real and imaginary motor execution, sensor level

## Abstract

Sensor-level human brain activity is studied during real and imaginary motor execution using functional near-infrared spectroscopy (fNIRS). Blood oxygenation and deoxygenation spatial dynamics exhibit pronounced hemispheric lateralization when performing motor tasks with the left and right hands. This fact allowed us to reveal biomarkers of hemodynamical response of the motor cortex on the motor execution, and use them for designing a sensing method for classification of the type of movement. The recognition accuracy of real movements is close to 100%, while the classification accuracy of imaginary movements is lower but quite high (at the level of 90%). The advantage of the proposed method is its ability to classify real and imaginary movements with sufficiently high efficiency without the need for recalculating parameters. The proposed system can serve as a sensor of motor activity to be used for neurorehabilitation after severe brain injuries, including traumas and strokes.

## 1. Introduction

A study of neurophysiological brain activity attracts significant interest of researchers from various fields of science due to its interdisciplinary nature. The considerable progress in the development of experimental approaches for neuroimaging and mathematical methods for big data analysis provides great opportunities for vast and detailed studies of specific phenomena in the brain neural network and the creation of sensor systems for monitoring brain states. Recent progress in this field has been achieved at the junction of mathematics, physics, engineering and neuroscience [1]. This is confirmed by an increasing number of papers related to brain research and published in multidisciplinary journals (see, e.g., [2,3,4,5]).

From the viewpoint of nonlinear dynamics, brain is a very complex dynamical system, containing around 86 billion neurons [6]. The neurons are connected by synapses, thus forming a complex network with nodes and links represented respectively by neurons and synapses. Specific features of time-spatial activity of the neural network in the brain cortex and cooperative dynamics of different brain areas (e.g., event-related synchronization/desynchronization) on a sensor level provide important information about the current state of the nervous system and cognitive brain ability [7,8,9,10]. Particular brain states are associated with motor brain activity during either real or imaginary movement.

Revealing specific features of spatial brain cortex activity related to real motions and motor imagery of different limbs can be essential not only for basic research in neuroscience, but also for applications in medicine to improve the quality of life of post-traumatic and post-stroke patients using brain-computer interfaces (BCI) for rehabilitation [11,12,13] or to control prostheses and exoskeletons [14]. One of the important BCI functions is online detection of specific features of electromagnetic brain activity using electroencephalography (EEG) [15] or magnetoencephalography (MEG) [16], and transformation of certain patterns into control commands to perform specific actions in the environment without the need of “classical” methods of human–machine interaction [17].

Apart from EEG and MEG, other methods are also used to acquire information about brain states. In particular, functional near-infrared spectroscopy (fNIRS) [18,19] is a powerful tool of noninvasive optical imaging successfully used in BCI for registration of brain activity and control command formation [20,21,22]. Control commands for this kind of BCI should not be affected by any muscular activity [23]. Therefore, a study of brain states related to motor imagery is very important for designing such BCI [16,24]. Motor imagery is a mental process by which a person rehearses or simulates a given action with no real motor activity. Some researchers treat motor imagery as a conscious application of unconscious preparation for real motor activity [25]. A number of studies have highlighted common features for real and imaginary motor activity [26,27,28]. One of the common features, important for the BCI development, is that the cortical layout in the primary motor cortex M1 is quite similar between motor execution and motor imagery.

The most popular technique for studying brain activity during motor executions is EEG, that implies the location of special sensors on the scalp (or directly into the brain) and recording EEG signals as electric currents generated by a group of neurons [29]. The EEG signal or electrical response of the neural network is characterized by a complex time-frequency structure containing specific frequency ranges, oscillatory patterns, stochastic components, artifacts, etc. [30]. It is well known that there is a strong correlation between EEG rhythmic activity and functional states of the organism [31,32,33] that can be used for revealing specific features related to real and imagery motor activity [34,35,36]. The problem of detection and classification of different types of motor execution requires the application of various methods for time-frequency and spatio-temporal analyses [37], including artificial intelligence methods [4], recurrence measures of signal complexity [38], as well as event-related synchronization (ERS) and event-related desynchronization (ERD) [39].

In this work, we use fNIRS, an efficient noninvasive technique for brain activity estimation [40], that employs near-infrared light to detect changes in oxygenated (HbO) and deoxygenated (HbR) hemoglobin levels due to hemodynamic brain activity and the rapid delivery of oxygenated blood to active cortical areas through neurovascular coupling [41]. A high efficiency of fNIRS is achieved due to the use of laser lights with two different wavelengths which penetrate most tissues in the head, but are highly absorbed by oxyhemoglobin (HbO) and deoxyhemoglobin (HbR). According to changes in the absorption of these two wavelengths, corresponding changes in HbO and HbR concentrations can be calculated and hence the oxygenation of brain tissues can be assessed.

It should be noted that fNIRS has a common physiological basis with functional magnetic resonance imaging (fMRI), so that their signals are correlated [42]. Meanwhile, fNIRS has many advantages over fMRI, namely, resistance to motor artifacts, high mobility due to significantly smaller sizes, and higher temporal resolution for the determination of oxyhemoglobin and deoxyhemoglobin. At the same time, fNIRS has some drawbacks, in particular, fNIRS has lower spatial resolution than fMRI. Despite relatively low temporal resolution and the existence of time delay in the hemodynamic response compared to EEG signals, fNIRS is a powerful tool of functional neuroimaging and can compete with other imaging techniques such as fMRI and EEG. The fNIRS imaging is superior to other techniques in studying neural activity in the primary motor cortex M1 because this area lies on the outer cortices which is within the blood oxygenation level deprivation (BOLD) scanning range for fNIRS [43]. A significant advantage of fNIRS over MEG is its portability and more easy mounting of sensors on the patient’s head. Moreover, the fNIRS equipment is cheaper as compared to fMRI and MEG. Thus, fNIRS holds a special place among modern methods of neuroimaging due to a good combination of temporal and spatial resolutions, overall mobility and simplicity of use.

In this paper, we analyse fNIRS data acquired during human real and imaginary motor activity associated with motor cortex hemodynamics. In particular, we extract specific features of the fNIRS signals related to different types of motor activity, which can be used in BCIs. We also propose an universal method to classify fNIRS trials obtained during motor imagery and develop a sensor of motor activity to be used for neurorehabilitation systems after various brain injuries, including stroke.

## 2. Materials and Methods

### 2.1. Participants

Twelve healthy volunteers (age: 22–38 years, gender: 7 men and 5 women), right-handed, amateur practitioners of physical exercises, non-smokers participated in the experiment. None of the subjects had diagnosed diseases of the musculoskeletal system and neurological diseases and did not take medications. Every participant was asked to maintain a healthy lifestyle with 8-hours night rest for 48 h before starting the experiment.

Each participant provided informed written consent before participating in the experiment. The experimental procedure was performed in accordance with the Helsinki’s Declaration and approved by the local Ethics Committee of the Innopolis University.

### 2.2. Experimental Equipment

The experiment was designed to record a hemodynamic neuronal response in the motor cortex using fNIRS which records fast changes in the brain activity. The fNIRS signals were acquired by the NIRScout device manufactured by the NIRx Company (Germany). The NIRScout system has a 7.8125-Hz resolution and contains 8 sources and 8 detectors placed on the subject’s scalp in the primary motor cortex area (M1) as shown in Figure 1a. Each pair “source–detector” was placed close enough to each other (about 3 cm) to form a fNIRS channel. In our experiments we used 20 fNIRS channels and 9 EEG electrodes located according to the international scheme “10–10” scheme [44], as illustrated in Figure 1b.

All experiments were carried out in the Neuroscience and Cognitive Technology Lab of the Innopolis University.

### 2.3. Experimental Procedure

The experiment was performed as follows. The subjects were sitting on a comfortable chair while performing motor actions or motor imaginary of left and right hands according to the corresponding text command on a computer monitor placed in front of the subject’s eyes at a distance of 70–80 cm.

The experimental design is schematically shown in Figure 2. Each experiment began and ended with a 3-min recording of background brain activity, during which the subject were instructed to relax and make no hand movements. The experiment included two sessions (Figure 2a). In the first session, the subject was asked to perform real movements with left or right hand according to the corresponding command on the screen. Then, after a short break, in the second session the subject was asked to imagine the same type of movement according to the corresponding command on the screen. Each fNIRS trial during every session consisted of the text command presentation indicating the type of motor activity (the subject was given 15 s to perform required movement) and the rest interval (15 s from the end of motor activity till the next command). There were 10 trials for each type of motor activity.

Hand movement consisted of repeated bending/unbending of fingers to the center of the palm (similar to clenching of imaginary ball), as illustrated in (Figure 2b). The repeated movements were performed at the pace comfortable for the subject.

### 2.4. Data Acquisition and Pre-Processing

In the fNIRS experiments, we used a laser light with two wavelengths, $\lambda_{1}=785$ nm and $\lambda_{2}\;=\;850$ nm, that can pass through skin, bone and water, but are highly absorbed by oxyhemoglobin and deoxyhemoglobin, respectively [45]. In the developed configuration for the fNIRS recording, the light sources and detectors were placed on the scalp and dual-wavelength light was transmitted through skin, skull, and top layer of the cerebral cortex.

As seen from Figure 1c, near-infrared light travels from the source to the detector through a specific path. First, it goes from the source to the tissue and at the depth of approximately 3 cm is reflected towards the detector. The path shape limits the distance between the source and detector to 2–3 cm and allows one to detect changes in the reflected near-infrared light.

In order to obtain information about changes in oxygenation of the tissue, we analyzed raw fNIRS data using special software. Since oxyhemoglobin and deoxyhemoglobin have different light absorption properties, we calculated changes in the reflected dual-wavelength light using the modified Beer–Lambert law [46]. For this purpose, we introduced a new characteristic measure (*H*) which reflects relative changes in oxyhemoglobin and deoxyhemoglobin.

The fNIRS data acquisition and pre-processing procedure were performed with software NIRScout. It is well-known that experimental fNIRS data are often affected by side physiological noises and artifacts, whose characteristic frequencies are in the fNIRS frequency band, including Mayer wave (with a typical frequency close to 0.1 Hz), respiration (close to 0.25 Hz), and heartbeat (close to 1 Hz). As was mentioned in the review paper [47], in many cases the band-pass filtering is mostly sufficient for removing low-frequency physiological noise in fNIRS data. According to this observation, we also applied the 0.01–0.1 Hz band-pass filter to the fNIRS signals using NIRScout to prevent the effect of side physiological activities.

The NIRScout software was also used for presentation of stimuli (text commands) and for logging the events (such as beginning and end of motor activity) in the protocol file. According to this file, 35-s long trials were formed by 5-s preparation before the text command, 15-s motor activity, and 15-s rest interval as illustrated in Figure 2b. The 5-s interval at the beginning of each trial was used for baseline correction. Specifically, the distribution of HbO/HbR was averaged over these 5 s and the obtained value was subtracted from the corresponding trial.

### 2.5. Data Analysis

The characteristic value $H^{i}$ ($i=1,\cdots ,N_{fNIRS}$, where $N_{fNIRS}=20$ is the number of fNIRS channels) provides information about HbO/HbR dynamics obtained from corresponding fNIRS channels. At the same time, to compare significance of such dynamics across the channels we need to introduce new characteristic $\delta H^{i,j}$ that can be calculated as follows.

First, we calculate the value of ${\langle H^{i}\rangle }_{T}$ as $H^{i}$ averaged for each fNIRS channel, separately for HbO and HbR, across time interval T∈(5,20) s corresponding to real or imaginary motor activity:(1)$${\langle H^{i}\rangle }_{T}=\int_{T}\Delta H^{i}\;dt.$$

In the previous papers [48,49], we have introduced the measure of connectivity based on the reconstruction of functional links between neuronal ensembles in different frequency bands by comparing spectral components of the EEG signals belonging to these bands. Here, we extend this approach to the time domain for analyzing the restoration of connectivity by similarity of hemodynamic responses in different areas of the motor cortex. This allows us to identify the cortical region in M1 with most similar activity for further classification of motor execution events.

According to our approach, we calculate matrices $\delta H^{i,j}$ of the difference between ${\langle H\rangle }_{T}$ for all fNIRS channels *i* and *j* (i,j=1,⋯,20) for each type of motor activity (real and imaginary) for both HbO and HbR as
(2)$$dH^{i,j}={\langle H^{i}\rangle }_{T}-{\langle H^{j}\rangle }_{T}.$$

Previous fMRI studies [50] indicate that the activation of primary motor cortex during real movement increases the level of oxyhemoglobin ($H_{\mathrm{HbO}}$) and decreases deoxyhemoglobin ($H_{\mathrm{HbR}}$). Therefore, in the resulting matrices $dH^{i,j}$ we leave only values $dH^{i,j}>0$ for HbO and $dH^{i,j}<0$ for HbR and take their absolute values for easier comparison. Finally, we construct the distributions $N(dH^{i,j})$ of $dH^{i,j}$ for each 20×20 matrix obtained for HBO and HbR in the case of left/right real and imaginary movements. These diagrams are present in Figure 3 for subject #3.

Based on the obtained distributions $N(dH^{i,j})$ we introduce cumulative distribution functions $F_{dH^{i,j}}\left(h\right)=P\left(dH^{i,j}\leq h\right)$ which yield the probability for $dH^{i,j}$ to be smaller than *h*. The fourth quartile of the F(h) distribution is the value of h≥0.75. The corresponding cumulative distributions are shown in Figure 3 by blue curves, while dashed vertical lines indicate the border of the fourth quartile. As a consequence, we only consider $d{\bar{H}}^{i,j}$ values that fall into the fourth quartile of distributions ($F({\bar{H}}^{i,j})\geq 0.75$) because the resulting values of $d{\bar{H}}^{i,j}$ are the most significant and thus can be used to find fNIRS channels suitable for the classifier (see Results). Thus, for further mathematical analysis we obtain the following matrixes of functional connectivity between different fNIRS channels corresponding to the dynamics of oxygenation and deoxygenation, respectively: (3)$$\begin{array}{cc} \delta H_{\mathrm{HbO}}^{i,j}= & \left\{\begin{array}{cc} |dH_{\mathrm{HbO}}^{i,j}|,\; & \mathrm{if}\;\;dH_{\mathrm{HbO}}^{i,j}>0\;\wedge \;F\left(dH_{\mathrm{HbO}}^{i,j}\right)\geq 0.75,\\ 0,\; & \mathrm{otherwise}, \end{array}\right. \end{array}$$(4)$$\begin{array}{cc} \delta H_{\mathrm{HbR}}^{i,j}= & \left\{\begin{array}{cc} |dH_{\mathrm{HbR}}^{i,j}|,\; & \mathrm{if}\;\;dH_{\mathrm{HbR}}^{i,j}<0\;\wedge \;F(|dH_{\mathrm{HbR}}^{i,j}\left|\right)\geq 0.75,\\ 0,\; & \mathrm{otherwise}. \end{array}\right. \end{array}$$

### 2.6. Algorithm for Classification of Brain Activity During Real and Imagery Motor Executions

Let us now consider the online algorithm based on the construction of a decision tree for binary classification of brain activity during real and imagery motor executions. The tree-like model of decisions was created as a result of empirical analysis of the experimental data. The proposed algorithm for processing fNIRS data is illustrated in Figure 4 in the form of a flowchat which contains the following main steps.

For each considered channel *i* and type of motor activity (right/left hand, execution/imagery) we subtract spatial oxyhemoglobin (HbO) ($H_{\mathrm{HbO}}^{i}$) and deoxyhemoglobin (HbR) ($H_{\mathrm{HbR}}^{i}$) distributions for the right hemisphere ($H_{R}^{j}$, fNIRS channels *j* of interest from right hemisphere) from the corresponding distribution for the left hemisphere ($H_{L}^{i}$, symmetrical channels *i* from left hemisphere). Similar to our approach (Equation 2) which uses average values, we calculate differences for individual symmetrical channels in the left and right hemispheres as
(5)$$\Delta H_{\mathrm{HbO}}^{i}=H_{\mathrm{HbO},\;L}^{i}-H_{\mathrm{HbO},\;R}^{j}\;,\;\;\Delta H_{\mathrm{HbR}}^{i}=H_{\mathrm{HbR},\;L}^{i}-H_{\mathrm{HbR},\;R}^{j}\;.$$Then, we average $\Delta H_{\mathrm{HbO}}^{i}$ and $\Delta H_{\mathrm{HbOR}}^{i}$ over the time interval corresponding to motor activity T∈(5,20) s to find ${\langle \Delta H_{\mathrm{HbO}}^{i}\rangle }_{T}$ and ${\langle \Delta H_{\mathrm{HbO}}^{i}\rangle }_{T}$ as
(6)$${\langle \Delta H_{\mathrm{HbO}}^{i}\rangle }_{T}=\int_{T}\Delta H_{\mathrm{HbO}}^{i}\;dt\;,\;\;{\langle \Delta H_{\mathrm{HbR}}^{i}\rangle }_{T}=\int_{T}\Delta H_{\mathrm{HbR}}^{i}\;dt.$$For each separate fNIRS signal trial, we calculate characteristics $C_{R}$ and $C_{L}$ taking into account the following criteria for each considered symmetric fNIRS channels in the left and right hemispheres.(i)If ${\langle \Delta H_{\mathrm{HbO}}^{i}\rangle }_{T}>0$ and ${\langle \Delta H_{\mathrm{HbR}}^{i}\rangle }_{T}<0$ is true for one of the channels *i*, then $C_{R}:=C_{R}+1$ (value $C_{R}$ takes discrete values, minimal value is $C_{R}=0$ and peak value is equal to the number of considered fNIRS channels of interest in one of the hemispheres).(ii)If ${\langle \Delta H_{\mathrm{HbO}}^{i}\rangle }_{T}<0$ and ${\langle \Delta H_{\mathrm{HbR}}^{i}\rangle }_{T}>0$ is true for one of the channels *i*, then $C_{L}:=C_{L}+1$ (value $C_{R}$ takes discrete values, minimal value is $C_{L}=0$ and peak value is the same as peak value of $C_{R}$).Finally, we make a decision according to the following criteria.(i)If $C_{R}>C_{L}$, then right-hand (real or imaginary) motor activity takes place.(ii)If $C_{R}<C_{L}$, then left-hand (real or imaginary) activity takes place.(iii)If $C_{R}=C_{L}$, then the type of activity is uncertain.

### 2.7. Estimation of Classification Accuracy of Brain Activity During Motor Execution and Motor Imagery

In order to generalize the results of statistical analysis to an independent data set, we applied the classical *k*-fold cross-validation technique [51] with k=10 and 20 fNIRS trials to every subsample. Additionally, the accuracy of the proposed detection method was evaluated by computing a percentage of true positive and false positive, true and false negative detections, sensitivity and specificity [52,53] using a 20-min experimental session, during which (i) the commands described in Section 2.3 (to perform real/imaginary movement with left/right hand) were issued to the subject and (ii) the classification algorithm recognised the type of movement using only 6 fNIRS channels. The true positive (TP) was computed as a percentage of correctly classified trials of right hand, and true negatives (TN) correctly classified trials of left hand. The false positive/negative (FP/FN) represents a percentage of incorrect automatically detected motor actions (FP being the trials classified as right-hand motor activity, but actually performed as left-hand motor activity, while FN denotes the opposite situation).

The true positive fraction (TPF) or sensitivity, true negative fraction (TNF) or specificity, and false positive fraction (FPF) of the method were assessed as [54]
(7)$$TPF=\frac{TP}{TP+FN}\times 100\%,\;TNF=\frac{TN}{TN+FP}\times 100\%,\;FPF=1-TNF.$$

## 3. Results

### 3.1. Spatial Brain Activity During Motor Execution and Motor Imagery

Figure 5 displays the results of the analysis of complex spatial brain activity in the primary motor cortex M1 detected by fNIRS. In Figure 5a,b we plot spatial distributions of oxyhemoglobin and deoxyhemoglobin, averaged over 10 trials of right-hand motor execution and right-hand motor imagery, respectively. The results for all 20 fNIRS channels are presented as mean values with a standard error.

In the case of motor execution, as seen from Figure 5a, several fNIRS channels demonstrate very specific dynamics; increasing oxyhemoglobin is accompanied by a corresponding decrease in deoxyhemoglobin during any motor activity, either real or imaginary (in time intervals between vertical dashed lines). We should note that this dynamics is much more pronounced in the left hemisphere (e.g., for channels 2, 3, 7, 8) for right hand and in the right hemisphere (for symmetric channels 12, 13, 17, 18) for left hand. Such a behavior, strongly linked to specific fNIRs channels, opens the opportunity to classify motor activity according to right and left hands.

It is worth noting that the above dynamics is less notable for motor imagery than for motor execution. As seen from Figure 5b, the oxyhemoglobin level slowly increases approximately 10 s after the beginning of movement imagination. The presence of the well-pronounced features of the hemodynamic response during motor imagery is very promissing for using fNIRS signals, to some degree, for detection of imaginary movements.

The most informative channels are those which display pronounced features of the hemodynamic response when performing both motor execution and motor imagery with left or right hand. The analysis of these channels is present in Figure 6 where we plot the matrices of functional connectivity (Equation 3) and () in the motor cortex for different types of motor activity (real/imaginary) related to right and left hands. First of all, one can clearly seen laterality between real and imaginary movements of left and right hands. When moving the right hand, the channels 2, 7 and 8 in the left hemisphere are most active exhibiting largest changes in the hemodynamic response. Whereas, when moving the right hand, the symmetric channels 12, 17 and 18 in the right hemisphere display the most pronounced activity. As a consequence, both an increase in oxygenation and a decrease in deoxygenation turn out to be informative in the mentioned channels, and its dynamics and connections with other channels are clearly seen in the connectivity matrices distributions. Secondly, the comparison of the brain response to real and imaginary movements allows us to conclude that motor execution yields a more accurate hemodynamic response. However, in the case of motor imagery, there is a strong connection with channel 2 in the left hemisphere for right hand, and channels 11 and 12 in the right hemisphere for left hand. At the same time, there is the activation (albeit less in amplitude as compared to real movements) of channels 7/8 and 17/18 for left/right hand, respectively. Since the particular areas of the motor cortex are responsible for different types of movement (real/imaginary), the fNIRS neuroimaging can be used for real-time diagnostics of motor activity.

### 3.2. Results of Real-Time Classification of Brain Activity

The main problem when creating any BCI based on motor imagery is a correct classification of single EEG/MEG trials. While averaged trials are usually exhibit a clearly pronounced difference between various types of movements (e.g., with left/right hand motor imagery), in the case of single trials, the classification problem is more drastic due to a high variability of EEG or MEG brain signals during imagination, as well as the existence of strong noise. Typically, the classification accuracy does not exceed 80% when special mathematical methods are applied, such as, e.g., SVM machines [55], wavelets [36,56], multilayer perceptrons [4], and recurrence quantitative measures [38].

The main advantage of fNIRS, as we demonstrated in the previous section, is a stable picture of spatial dynamics of oxygenation and deoxygenation in the M1 motor cortex during both real and imaginary movements. As seen from Figure 5, oxyhemoglobin and deoxyhemoglobin distributions exhibit qualitatively different dynamics on fNIRS channels in the left hemisphere and symmetric channels in the right hemisphere during 5–20 s intervals of right-hand motor execution/imagery. Considering the difference between hemoglobin dynamics in the left and right hemispheres, we can conclude that the oxyhemoglobin difference is mostly positive, while the deoxyhemoglobin difference is typically negative. In the case of left-hand motor activity, the opposite situation occurs, namely, the difference between signals from the left and right hemispheres is negative for oxyhemoglobin and positive for deoxyhemoglobin. It should be noted that these differences are manifested in the dynamics of i={2,7,8} fNIRS channels in the left hemisphere and j={12,17,18} symmetric channels in the right hemisphere.

This situation is illustrated in Figure 7, where plot the differences in temporal distributions of oxyhemoglobin and deoxyhemoglobin between left and right hemispheres for 10 trials recorded by fNIRS channels 2 and 12 for one subject. These results display the difference in the brain response between real and imaginary right-hand movements. The repeatability of the hemodynamic response in the motor cortex is clearly seen in Figure 7, when the same type of motor action is performed or imagined. One can also note the stability of the repeated maxima in the oxygenation difference between left and right hemispheres corresponding to real movement, whereas the maxima corresponding to motor imagery are unstable and varied over the trials. However, the tendency to increase the positive difference remains. Similar conclusions can be made when analyzing deoxygenation, but in this case the difference is negative. As a result, we can use these changes in oxygenation and deoxygenation dynamics in the left and right hemispheres as markers for classifying the type of movement and develop a method for online classification of right/left-hand motor activity based on fNIRS data.

Finally, we implemented the classification algorithm described in Section 2.6 in our fNIRS-based experimental system for online classification of real and imaginary motor actions. We used the proposed classifier with six fNIRS channels i={2,7,8} in the left hemisphere and j={12,17,18} symmetric channels in the right hemisphere. Notably, in the majority of cases the type of motor action (both real and imagery) in all subjects was correctly identified by the data from only three fNIRS channels. Table 1 shows the results of automatic classification of left/right hand real and imaginary movements, as well as statistical analysis of true positive fraction, true negative fraction, and false positive fraction calculated with Equation (Equation 7).

Figure 8 shows the receiver operating characteristic (ROC) curves for right/left hand motor action, which displays the values of false positive fraction (FPF) versus true positive fraction (TPF). In the considered case, the ROC curves have two segments (0,0)−(FPF,TPF)−(100,100) (in %) because a binary classifier (in our case, the decision tree) can only take the values of either 0 or 1. The statistical analysis shows that the selectivity and specificity of the classifier are sufficiently high and reach 85%–100%. The areas under the ROC curves in Figure 8 indicate that the accuracy of the binary algorithm for motor execution is higher than for motor imagery. The number of incorrectly detected and uncertain events is about 5%–20%. This reflects the fact that real and imaginary motor actions by left and right hands are reliably recognized using eight sources and eight fNIRS detectors, followed by data processing of spatial brain activity with the simple online decision tree-based classifier.

## 4. Discussion

The classification of brain activity trials associated with motor imagery using different methods of neuroimaging is a modern research topic widely explored by many researchers from different fields of science [13,57,58]. The diverse literature indicates that different brain regions can be involved in motor imaginary [59], including primary motor cortex (M1) [60], supplementary motor area (SMA) [61], posterior parietal cortex (PPC) [62], prefrontal cortex [63], etc. Moreover, there are at least two types of motor imagination (kinesthetic and visual) characterised by the involvement of various areas of the brain cortex. In particular, a well-pronounced suppression of alpha activity was observed in the occipital region in subjects exhibiting the visual type of motor imagery. In contrast, kinesthetic subjects displayed a pronounced suppression of mu activity in the motor and somatosensory cortexes [16]. In the referenced paper, 65%–80% classification accuracy was reported when choosing optimal MEG channels for both kinesthetic and visual untrained subjects. Notably, such accuracy for untrained subjects is similar to classification accuracy obtained in EEG studies with trained subjects [64]. Based on the effect of suppressing mu/alpha and beta activity during motor imagery, it is possible to obtain up to 70%–85% accuracy using various classifiers, including machine learning methods, SVM, ICA, etc. [65]. The main obstacle for achieving higher classification accuracy of single trials is the significant variability of temporal-spatial brain activity during motor imagery, which does not allow revealing universal patterns associated with a particular type of motor imagination.

In addition, the use of the hemodynamic response for the classification of motor imagery (e.g., for neurorehabilitation) also imposes a number of limitations. Firstly, the hemodynamic response is quite inertial, so that the registration of mild changes in oxygenation and deoxygenation of blood is required in the area of interest. Secondly, due to a relatively high cost and a large size, the MRI equipment is usually used in clinics for diagnosis, but not for regular training of motor imagery to restore motor activity after a stroke, as in the case of using EEG. In this context, it is of considerable interest to use the fNIRS device as a sensor of brain activity with an appropriate classification algorithm for the allocation of motor-related brain activity on the sensor level. As we have shown, the hemodynamic response generated by motor activity is strong enough to distinguish the corresponding region of the motor cortex responsible for the motor pattern formation. Thanks to the hemodynamic analysis, which is not sensitive to EEG background activity, it is possible to increase classification accuracy due to better signal repeatability in the analysis of single events. As a result, the classification accuracy of imaginary activity was increased up to more than 90%, which is quite sufficient for medical practice in neurorehabilitation procedures. It is important to emphasize that the fNIRS-based sensor is able to classify not only the kinesthetic mode of imagination, which is characterized by a pattern similar to real movements but also the visual type of imagination. However, we have to assume that in some cases, the classifier cannot recognize motor activity (the “uncertanity” algorithm branch in Figure 4).

In conclusion, it should be noted that imaginary movements are often used to form control commands in active brain-computer interfaces (BCI) [66,67]. Typically, such systems are based on the registration of electric brain activity (EEG) due to the fact that EEG gives a very fast response to changes in cognitive states. Moreover, EEG sensors are cheap and portable and do not impede the subject movement. Although the proposed fNIRS-based sensor system is also portable that allows the installation of sources and detectors faster and more convenient than EEG (or the same in the case of dry EEG electrodes [68,69,70]), the fNIRS device is more expensive than EEG. At the same time, the number of needed fNIRS channels for classifying activity is not high that can reduce the system cost. On the other hand, due to slow hemodynamic response, fNIRS exhibits a relatively large delay in detecting brain activity as compared to EEG, which restrict its application in BCIs. Nevertheless, the proposed fNIRS classifier can be used as a motor imagery sensor in neurorehabilitation systems with patients for which a 10–15 s latency is not significant. However, if such a BCI is used in game interfaces [71,72] or control interfaces of external devices (wheelchair, manipulator, etc.) [73,74,75], such delay time is critical and negate the advantages of our system in classification accuracy of brain states. One of the possible solutions to this problem could be the use of hybrid fNIRS-EEG BCIs [76,77,78]. The consideration of this type of classifier is out of the scope of this paper and might be the matter of future research.

## 5. Conclusions

In this paper, we have carried out the analysis of fNIRS data acquired during real and imaginary movements. Distinct spatial dynamics in the motor cortex when performing motor actions (real or imaginary) with the left or right hand exhibits pronounced laterality between two hemispheres. This allowed us to reveal hemodynamic biomarkers for classification of the type of movement. The proposed fNIRS-based sensor provides close to 100% recognition accuracy in the detection of real movements, while the classification accuracy of motor imagery is a little smaller and reach 90%.

The important advantage of the proposed method is the possibility to efficiently classify different types of movement, both real and imaginary, without recalculation of the system parameters. This essential feature of the developed sensor results from pronounced laterality of the hemodynamic brain response to motor activity.

The knowledge of the hemodynamic behavior in the motor cortex during real and imaginary motor activity along with approaches for its detection can be helpful not only for fundamental studies on human motor-related tasks but also for the development of fNIRS-based BCIs.

## Figures and Tables

**Figure 1 sensors-20-02362-f001:**
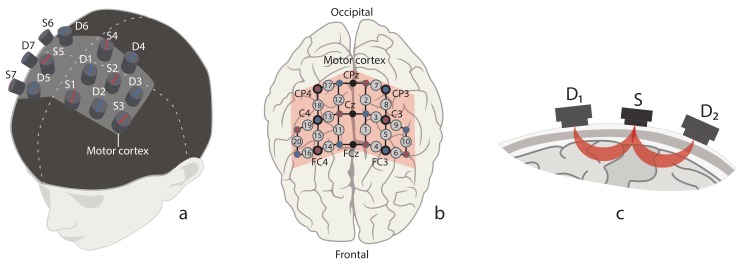
(**a**) Location of fNIRS sources (marked with red) and detectors (marked with blue) on the subject’s head in the area of primary motor cortex (shaded area). (**b**) Location of 20 fNIRS channels (grey circles with channel numbers) across motor cortex (shaded area) and 9 EEG electrodes (CP4, CPz, CP3, C4, Cz, C3, FC4, FCz, FC3) according to “10–10” scheme (black circles with channel names). (**c**) Schematic illustration of traveling path of near-infrared light from source (*S*) to neighbour detectors ($D_{1}$ and $D_{2}$) through brain cortex matter.

**Figure 2 sensors-20-02362-f002:**
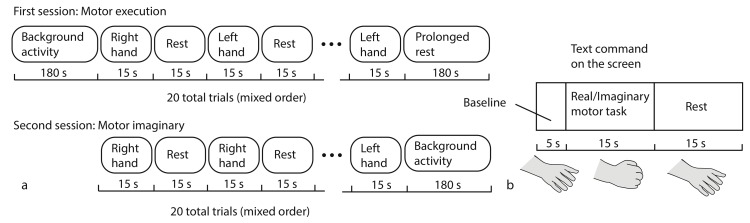
Schematic representation of (**a**) experimental design and (**b**) task execution during a long fNIRS trial.

**Figure 3 sensors-20-02362-f003:**
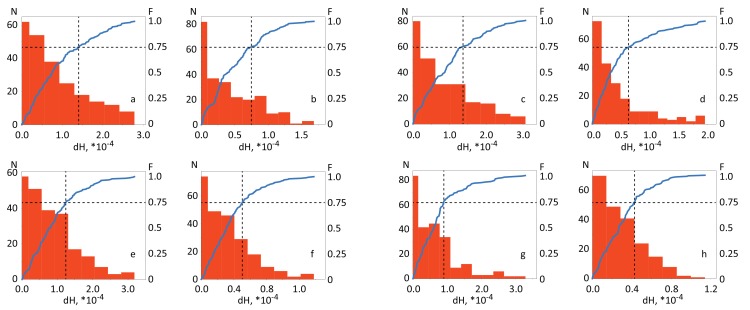
Distribution of $|dH^{i,j}|$ values (Equation 3) and cumulative distribution function *F* for different types of motor activity. (Upper line) Real movement: (**a**) HbO and (**b**) NbR for right hand and (**c**) HbO and (**d**) NbR for left hand. (Lower line) Motor imagery: (**e**) HbO and (**f**) HbR for right hand and (**g**) HbO and (**h**) HbR for left hand. The vertical dashed line indicates the border of the fourth quartile (75%) of the distributions. These results were obtained for subject #3.

**Figure 4 sensors-20-02362-f004:**
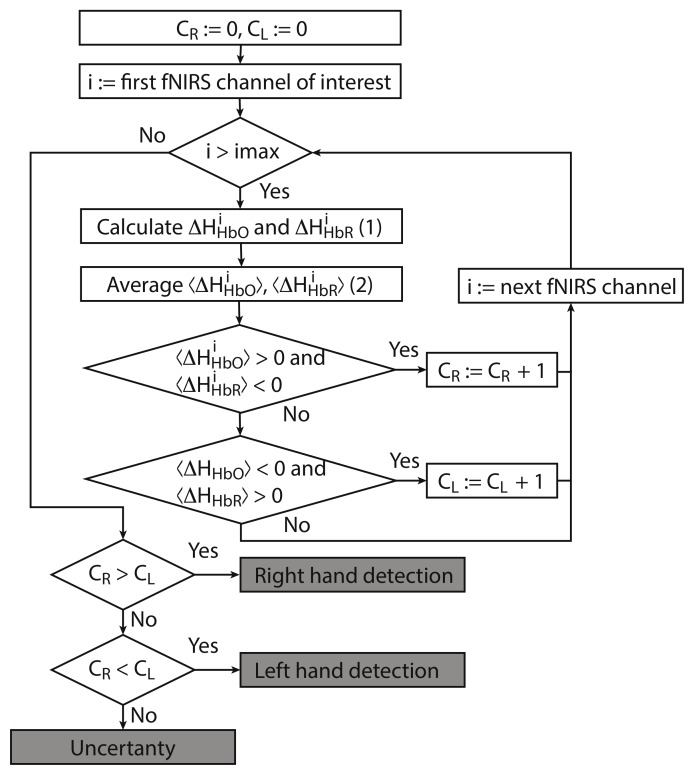
Flowchart of online classification algorithm for fNIRS signals corresponding to left/right-hand movement. The main advantage of the algorithm is that it can be used for both real and imaginary movements.

**Figure 5 sensors-20-02362-f005:**
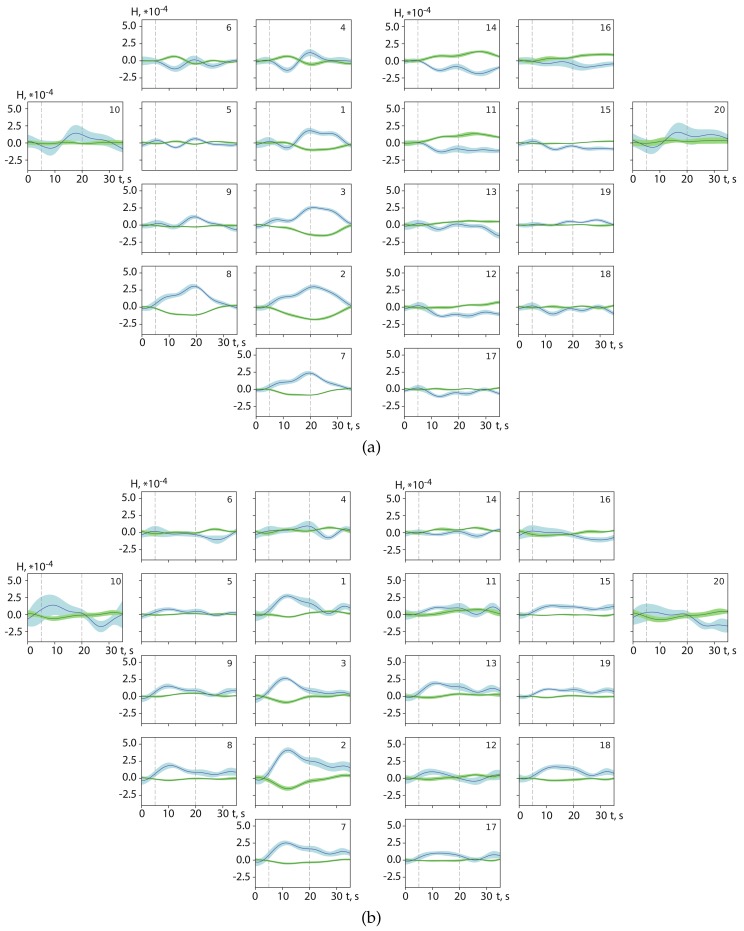
Distributions (mean ± S.D.) of oxyhemoglobin (blue) and deoxyhemoglobin (green) for (**a**) motor execution and (**b**) motor imaginary of right hand, for all 20 fNIRS channels. The data are averaged over 10 trials obtained for one subject. Vertical dashed lines indicate the beginning and end of each movement. The channel number is indicated in the upper right corner of each distribution. The results were obtained for subject #3.

**Figure 6 sensors-20-02362-f006:**
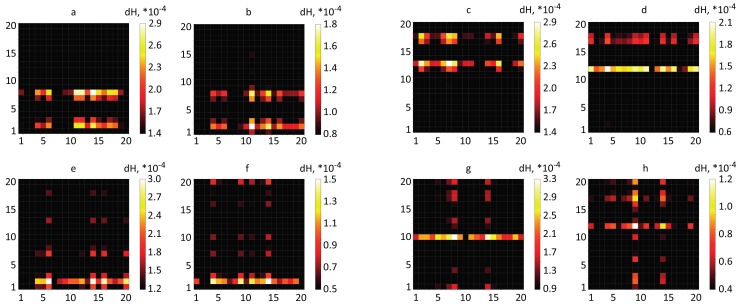
Distributions of δH given by (Equation 3) and () across fNIRS channels for different types of motor activity. (Upper line) Motor execution: (**a**) HbO and (**b**) HbR for right hand and (**c**) HbO and (**d**) HbR for left hand. (Lower line) Motor imagery: (**e**) HbO and (**f**) HbR for right hand and (**g**) HbO and (**h**) HbR for left hand. The results were obtained for subject #3.

**Figure 7 sensors-20-02362-f007:**
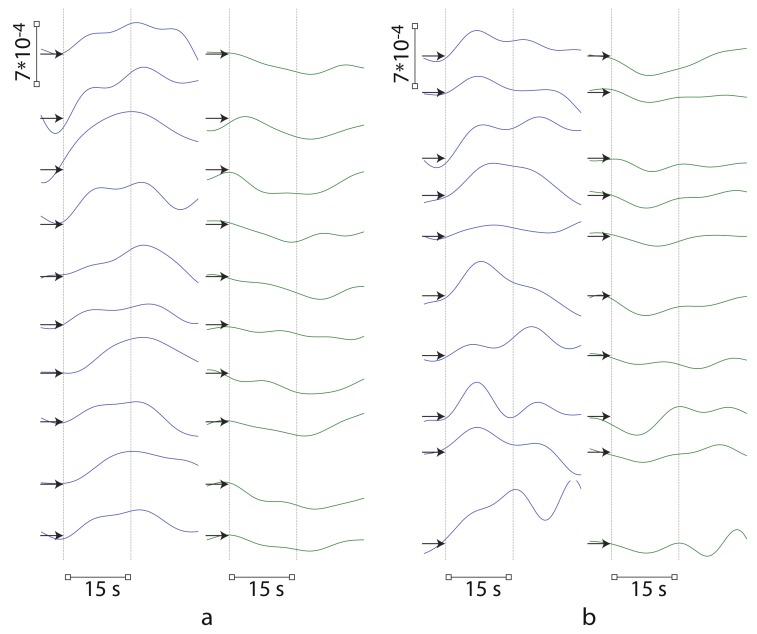
Differences in oxyhemoglobin (blue) and deoxyhemoglobin (green) distributions between left and right hemispheres. The results are present for fNIRS channel 2 for 10 trials corresponding to (**a**) real and (**b**) imaginary right-hand movements for subject #3. The vertical dashed lines indicate the time moments of the beginning and end of the movements. The arrows mark zero level in each distribution.

**Figure 8 sensors-20-02362-f008:**
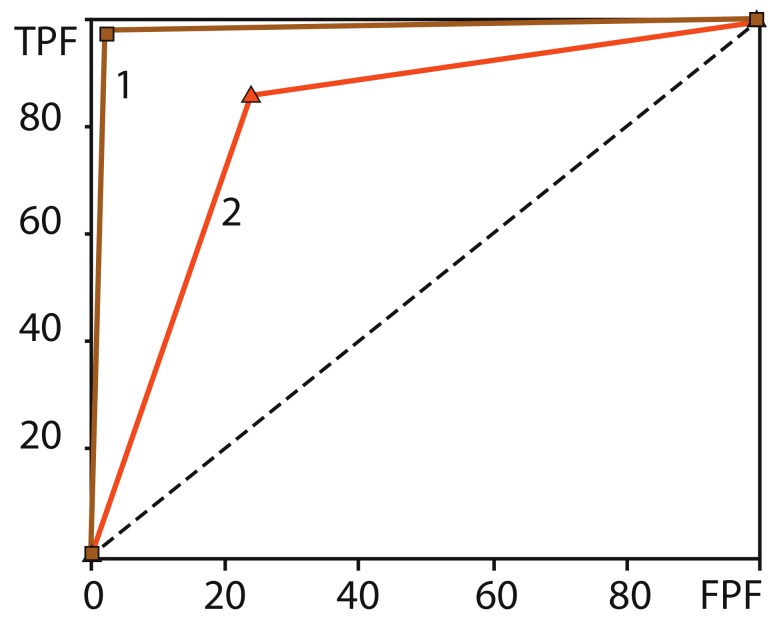
Receiver operating characteristic (ROC) in the units of true positive fraction (TPF) versus false positive fraction (FPF) [in %] of binary classification for motor execution (brown curve 1) and motor imaginary (orange curve 2).

**Table 1 sensors-20-02362-t001:** The results of automatic classification of different types of motor action (mean ± S.D.) using i={2,7,8} and j={12,17,18} fNIRS channels in the left and right hemisphere, respectively.

Movement	Automatic Detection				TPF, %	TNF, %	FPF, %
	TP	FP	TN	FN			
Real	99.0±3.1	2.0±4.2	98.0±4.2	2.0±4.2	98.2±3.8	98.0±4.2	2.0±4.2
Imaginary	86.0±10.7	25.0±12.6	78.0±9.1	14.0±8.4	86.1±7.9	76.2±11.2	23.8±11.2

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
