# Peer review of "Functional Near-Infrared Spectroscopy for the Classification of Motor-Related Brain Activity on the Sensor-Level"

_sensors, 2020, doi:10.3390/s20082362_

Round 1

Reviewer 1 Report

This manuscript proposed a novel method for fNIRS data classification. The data was collected from a real and a imaginary hand movement tasks. The authors were able to show that their proposed classification method can not only discriminate right-left side hand movement in both scenarios at a high classification accuracy.  The manuscript is well written, however, the following issues need to be addressed before it can be published:

  1. Introduction section, line 62 - 67, please provide more details about fNIRS technology, including specifically, what additional information about brain activity fNIRS can obtain compared with other neuroimaging techniques.
  2. Method section, line 76, please define what is conditionally healthy volunteers.
  3. Method section, line 84-87, these information should go to introduction section.
  4. Method section, line 91, please double check the source-detector pair distance of 4 cm. To the reviewer, this distance is rare. Most fNIRS studies used 2-3 cm as source-detector pair distance. Do you have a specific reason to pick up 4 cm?
  5. Method section 2.3, it is suggested to include a figure for the experimental procedure.
  6. There is no localization process motioned in the method section, please describe it. Otherwise, how would one know that where was the data collected from?
  7. Method section 2.5, line 145 and line 147, there were two instances of unclear notation dH^i_j. Please correct otherwise redefine them.
  8. Results section, line 190, there was a highlighted (of what?) in the text, please correct.
  9. Section 3.2 needs to go to the method section, in my opinion.
  10. I suggest the authors to include ROC curves in the result section to make the classification performance more clear to the readers.

Author Response

We thank both Reviewers for the careful reading of our manuscript and for their useful comments, which we considered in the revised version of the paper.

All changes in the revised manuscript are marked in blue except minor corrections (including language).

Remark 1: Introduction section, line 62 - 67, please provide more details about fNIRS technology, including specifically, what additional information about brain activity fNIRS can obtain compared with other neuroimaging techniques.

Our reply:

We have added additional information about fNIRS technology in the Introduction. We rewrote this part of the text to emphasize the choice of fNIRS as method for our research It should be noted that every neuroimaging technique has its own specific features, advantages and drawbacks. One of the important advantages of fNIRS imaging is its usage for studying the primary motor cortex M1 because this area is on the outer cortices within the blood oxygenation level deprivation (BOLD) scanning range for fNIRS.  

Remark 2: Method section, line 76, please define what is conditionally healthy volunteers.

Our reply:

We clarified the term “conditionally healthy volunteers” by adding the following sentence in the Section 2.1 of the manuscript: “None of the subjects had diagnosed diseases of the musculoskeletal system and neurological diseases and did not take medications.”

Remark 3: Method section, line 84-87, these information should go to introduction section.

Our reply:

Thank you for this comment. We have moved this phrase in Introduction (the penultimate paragraph) and extended it according to Comment 1.

Remark 4: Method section, line 91, please double check the source-detector pair distance of 4 cm. To the reviewer, this distance is rare. Most fNIRS studies used 2-3 cm as source-detector pair distance. Do you have a specific reason to pick up 4 cm?

Our reply:

Thank you for this observation. Indeed, the source-detector pair distance was 2-3 cm, as in the most fNIRS-based studies. This mistake was caused by the NIRScout software which indicates distances with respect to the sphericity of the subject’s head and the actual size of sensors. In our experiment we used a special cap with sockets for sensors and each pair of sensors was connected by a physical stabilizing link with a length of 2-3 cm to limit the distances between sources and detectors. We have corrected this mistake in the text:

line 103              “about 4 cm” changed to “about 3 cm”

line 133            “0.035–0.0045 m” changed to “2-3 cm”

Remark 5: Method section 2.3, it is suggested to include a figure for the experimental procedure.

Our reply:

The schematic representation of the experimental procedure has been included in the revised version of the manuscript as figure 2.

Remark 6: There is no localization process motioned in the method section, please describe it. Otherwise, how would one know that where was the data collected from?

We have revised Fig. 1b and added the location of few EEG electrodes according to the “10-10” scheme. This basic information about the location of the fNIRS channels will be useful for the reader. The corresponding changes have also been made in the caption of Fig. 1 and in the text.

Remark 7: Method section 2.5, line 145 and line 147, there were two instances of unclear notation dH^i_j. Please correct otherwise redefine them.

We have corrected these typos.

Remark 8: Results section, line 190, there was a highlighted (of what?) in the text, please correct.

We have removed this extra text.

Remark 9: Section 3.2 needs to go to the method section, in my opinion.

We have moved the part of Section 3.2 to the Method section. We have also added the new section 2.7 “Estimation of accuracy of classification of brain activity during real and imagery motor executions” where we included a part of the text from Section 3.2.

Remark 10: I suggest the authors to include ROC curves in the result section to make the classification performance more clear to the readers.

We have included the ROC curves of binary classification for motor execution and motor imagery (see Fig. 8 in the revised manuscript). In the considered case, the ROC curves have a piecewise linear form due to the fact that the ROC curve of the binary classifier (in our case, the decision tree) issuing 0 or 1 looks like two segments (0, 0) − (FPF, TPF) − (100, 100)%.

Reviewer 2 Report

  1. Line 190 : (of What?) is in red. Where this comes from? Delete it.
  2. What classifier is used in the work? Present the classifier and how its parameters are tuned/fixed. Justify the choice of the classifier.
  3. What type of cross validation is applied to the data.
  4. How data is pre-processed against presence of noise?

Author Response

We thank both Reviewers for the careful reading of our manuscript and for their useful comments, which we considered in the revised version of the paper.

All changes in the revised manuscript are marked in blue except minor corrections (including language).

Remark 1:  Line 190 : (of What?) is in red. Where this comes from? Delete it.

We have removed this extra text.

Remark 2: What classifier is used in the work? Present the classifier and how its parameters are tuned/fixed. Justify the choice of the classifier

In the paper, we propose and test a binary classifier based on the construction of a decision tree. The construction is based on our results on changes of blood oxygenation and deoxygenation which exhibit pronounced hemispheric lateralization when performing motor tasks with left and right hands. This fact allowed us to reveal biomarkers of hemodynamical response of the motor cortex on the motor execution, and use them for designing a sensing method for classification of the type of movement. The classifier parameters are the analyzed fNIRS channels. We have shown that for the examined group of healthy subjects the same 6 channels {2,7,8,12,17,18} located symmetrically in the left and right hemispheres are most optimally using. We have added relevant explanations in Section 2.6 “The algorithm for classification of brain activity during real and imagery motor”. To improve the article structure, according to the comments of the First Reviewer, we moved this subsection to Section “Materials and methods”.

Remark 3: What type of cross validation is applied to the data.

We have applied the classical k-fold cross-validation with k= 10 and 20 trials in every subsample. The corresponding description has been added in Section 2.7 “Estimation of accuracy of classification of brain activity during real and imagery motor executions”.

Remark 4: How data is pre-processed against presence of noise?

It is well-known that experimental data such as fNIRS is often affected by physiological noises and artifacts, such as Mayer wave (about 0.1 Hz), respiration (about 0.25 Hz), and heartbeat (about 1 Hz). Some researchers (e.g., Naseer N., Hong K. S. fNIRS-based brain-computer interfaces: a review, Frontiers in Human Neuroscience, 2015) suggest that in many cases band-pass filtering is mostly sufficient for removing physiological noise from fNIRS data. In our study we band-pass filtered fNIRS data in the range of 0.01–0.1 Hz. We believe that this pre-processing is enough to remove the effect of Mayer wave, respiration, and heartbeat.

Additionally, fNIRS data contains extra-cortical (superficial) noises that occur when the light travels through the extra-cortical layers (i.e., scalp, skull, cerebrospinal fluid) prior to/after reaching the cortical layers (i.e., gray and white matter). The usual way to deal with this type of noise is to detect additional fNIRS data on short-separation (SS) channels (source-detector pair with a distance less than 1 cm) and to subtract SS-measured data from initial fNIRS data. However, this approach requires additional fNIRS optodes of special type, which are not available in our experimental setup.

The corresponding brief discussion has been added in Section 2.4 “Data acquisition and pre-processing”.